# Macromolecules Absorbed from Influenza Infection-Based Sera Modulate the Cellular Uptake of Polymeric Nanoparticles

**DOI:** 10.3390/biomimetics7040219

**Published:** 2022-11-30

**Authors:** Daniel Nierenberg, Orielyz Flores, David Fox, Yuen Yee Li Sip, Caroline M. Finn, Heba Ghozlan, Amanda Cox, Melanie Coathup, Karl Kai McKinstry, Lei Zhai, Annette R. Khaled

**Affiliations:** 1Burnett School of Biomedical Sciences, College of Medicine, University of Central Florida, Orlando, FL 32827, USA; 2NanoScience Technology Science Center, University of Central Florida, Orlando, FL 32826, USA; 3Department of Chemistry, College of Science, University of Central Florida, Orlando, FL 32816, USA; 4Department of Materials Science and Engineering, College of Engineering and Computer Science, University of Central Florida, Orlando, FL 32816, USA; 5Biionix Cluster and Department of Internal Medicine, College of Medicine, University of Central Florida, Orlando, FL 32827, USA

**Keywords:** drug delivery, tumor, paclitaxel, protein corona, nanomedicine

## Abstract

Optimizing the biological identity of nanoparticles (NPs) for efficient tumor uptake remains challenging. The controlled formation of a protein corona on NPs through protein absorption from biofluids could favor a biological identity that enables tumor accumulation. To increase the diversity of proteins absorbed by NPs, sera derived from Influenza A virus (IAV)-infected mice were used to pre-coat NPs formed using a hyperbranched polyester polymer (HBPE-NPs). HBPE-NPs, encapsulating a tracking dye or cancer drug, were treated with sera from days 3–6 of IAV infection (VS3-6), and uptake of HBPE-NPs by breast cancer cells was examined. Cancer cells demonstrated better uptake of HBPE-NPs pre-treated with VS3-6 over polyethylene glycol (PEG)-HBPE-NPs, a standard NP surface modification. The uptake of VS5 pre-treated HBPE-NPs by monocytic cells (THP-1) was decreased over PEG-HBPE-NPs. VS5-treated HBPE-NPs delivered a cancer drug more efficiently and displayed better *in vivo* distribution over controls, remaining stable even after interacting with endothelial cells. Using a proteomics approach, proteins absorbed from sera-treated HBPE-NPs were identified, such as thrombospondin-1 (TSP-1), that could bind multiple cancer cell receptors. Our findings indicate that serum collected during an immune response to infection is a rich source of macromolecules that are absorbed by NPs and modulate their biological identity, achieving rationally designed uptake by targeted cell types.

## 1. Introduction

The critical interface between the surface of NPs and the surrounding environment is, in part, defined by the physicochemical characteristics of NPs, such as size, charge, or shape that influence the absorption of macromolecules from complex biofluids [1,2,3]. Early studies focused on decreasing the absorption of proteins by NPs, called biofouling, to improve *in vivo* performance [4]. Modifications of NPs, such as PEGylation (PEG), enhance the biocompatibility and immunoevasive properties of NPs. However, PEG may stimulate the production of anti-PEG antibodies that can induce severe allergic reactions [5]. PEG may also impede the cellular uptake and endosomal escape of particles—known as the “PEG dilemma” [6]. Thus, finding safer biomimetic alternatives for coating the surfaces of NPs that do not involve PEG is an unmet need. When NPs interact with proteins in biofluids, layers form termed the hard or soft protein coronas [7]. Hard protein coronas may remain relatively stable on NPs in circulation, while soft protein coronas could undergo a dynamic exchange of their components [8,9,10,11]. Several theories address this formation of a protein corona on NPs. The “Vroman effect” suggests that while the composition of absorbed proteins can change over time, the total amount of proteins remain stable [12]. Hence, higher abundance proteins initially absorbed could later be replaced with proteins of higher affinity. A more recent theory, based on data from label-free snapshot proteomics, posed that the quantity of proteins absorbed by NPs could change over time but not the composition [13]. Another factor to consider is that NPs themselves may dictate the absorption of proteins. Studies reported that the distribution of proteins found in the corona of NPs was distinct from the distribution found in the protein source. For example, the ten most abundant proteins in human blood serum represent ~86% of the total sera protein content but may only comprise 8–13% of the coronal proteins absorbed by NPs [14,15]. The critical intrinsic elements that influence corona development on NPs and contribute to cellular interactions remain to be identified. Elucidating these factors is necessary to enhance drug delivery to tumors mediated by NPs. 

Studies that define the characteristics of protein coronas formed on NPs often use normal (non-diseased) sera as a source for coronal proteins. Incubating NPs with normal sera enables insight into which protein components of blood may be influencing the biodistribution of NPs *in vivo* under native conditions, as has been shown for albumin [16] or apolipoproteins [17]. Incubating polymeric NPs, formed using a hyperbranched polyester (HBPE) polymer optimized by our lab [18,19], with normal mouse sera enhanced breast cancer cell uptake and improved delivery of anti-cancer drugs compared to PEGylated HBPE-NPs [19]. However, pre-treatment of HBPE-NPs with normal mouse sera did not reduce monocyte uptake (e.g., THP-1 cells). A challenge to improving circulation (and tumor accumulation) of systemically administered NPs is the uptake of particles by the mononuclear phagocytic system (MPS), also known as the reticuloendothelial system (RES). Reducing uptake by the MPS is especially important for polymeric NPs since we previously showed that a significant portion of these NPs when intravenously injected in mice, even when PEGylated, accumulate in the liver or spleen where mononuclear cells are abundant [20,21]. An alternative approach is the use of sera from diseased states that may contain a more complex collection of macromolecules for absorption by NPs. The formation of a protein corona on circulating NPs depends on the macromolecules found in sera, which can vary, as has been shown for particles treated with sera sourced from cancer or diabetes patients [22,23,24]. Hence the tumor uptake of NPs, like the HBPE-NPs, could be improved by pre-coating with sera that form a protein corona composed of a diversity of macromolecules that enhance the biological interface to favor particle circulation and cancer cell uptake.

During the innate and adaptive immune responses, antigen-activated leukocytes and lymphocytes traffic through the body to reach sites of infection or inflammation. In part, through proteolytic mechanisms that result in ectodomain shedding, adhesion molecules, cytokines, growth factors, enzymes, and other molecules are released into the blood. This process is significantly enhanced when immune cells are activated [25,26,27,28]. The resulting serum could be a rich and diverse source of proteins for absorption by NPs. To explore whether pre-treating HBPE-NPs with such a serum could enhance cancer cell uptake, we used sera from influenza A virus (IAV)-infected mice. The IAV infection runs a course from innate to adaptive responses over several days, enabling the daily collection of viral infection-derived sera (VS) that contain a diverse array of shed proteins. Using these sera, we pre-treated HBPE-NPs loaded with either a lipophilic tracking dye or a cancer drug and performed *in vitro* cell-based experiments and *in vivo* biodistribution studies to characterize the behavior of the HBPE-NPs. Sera from day 5 of the IAV infection (VS5) contained proteins absorbed by the HBPE-NPs that conferred improved tumor cell uptake and reduced monocytic cell uptake. This resulted in the accumulation of HBPE-NPs in tumors and reduced liver and spleen uptake. We identified the most abundant proteins absorbed by HBPE-NPs from VS3 and VS5 and noted differences in the protein profiles relevant to immune response-derived proteins and possible ligands for multiple cancer cell receptors. These results support that the protein corona formed on polymeric NPs, like the HBPE-NPs pre-treated with sera from the IAV infection, is composed of unique proteins that modulate cancer or immune cell uptake to optimize the biological identity of polymeric NPs for targeted accumulation in tumors. 

## 2. Materials and Methods

### 2.1. Nanoparticle Synthesis, Cargo Encapsulation, and PEG Functionalization

The HBPE polymer was synthesized following our previously published protocol [19]. To generate HBPE-NPs and encapsulate them with 1,1′-Dioctadecyl-3,3,3′,3′-tetramethylindocarbocyanine perchlorate (DiI) dye (Life Technologies, Thermo Fisher Scientific, Waltham, MA, USA) or 1,1′-Dioctadecyl- 3,3,3′,3′-tetramethylindotricarbocyanine iodide (DiR) dye (Life Technologies, Thermo Fisher Scientific, Waltham, MA, USA), 10 milligrams (mg) of HBPE polymer was mixed with 1 microgram (µg) of DiI or DiR dye and dissolved at 100 mg/milliliter (mL) in dimethyl sulfoxide (DMSO) (Thermo Fisher Scientific, Waltham, MA, USA). To load particles with paclitaxel (Taxol) (Cayman Chemical, Ann Arbor, MI, USA), 10 mg of HBPE polymer was combined with 2 mg of Taxol and solubilized in DMSO at 100 mg/mL. As previously described [19], the solvent diffusion method produced cargo-loaded HBPE-NPs with a carboxylated surface at a final concentration of 10 mg/mL. To PEGylate, the surface of the HBPE-NPs, PEG 2-aminoethyl ether acetic acid 10,000 molecular weight (MW) (MilliporeSigma, Burlington, MA, USA) was used. PEG was conjugated to HBPE-NPs using 1-ethyl-3-(3-dimethylaminopropyl) carbodiimide (EDC) (MilliporeSigma, Burlington, MA, USA) and N-hydroxysuccinimide (NHS) (MilliporeSigma, Burlington, MA, USA) chemistry, as described by Santra et al. [18]. Following the formation of carboxylated or PEGylated HBPE-NPs, unreacted chemicals and large particulates were removed following methods employed in Nierenberg et al. [19]. Methods for quantifying dye or drug loading encapsulated within HBPE-NPs are detailed in Nierenberg et al. [19].

### 2.2. IAV Infection and Recovery of Sera for Treating HBPE-NPs

For collecting serum, 2–3-month-old C57BL/6 female mice were intranasally treated with a lethal dose (30 LD_50_) of H3N2 A/Philippines/2/82/x-79 Influenza A virus in 50 μL PBS. Blood was collected from mice at 3, 4, 5, or 6 days post IAV infection by a terminal cardiac puncture while mice were under anesthesia using tribromoethanol (MilliporeSigma, Burlington, MA, USA). Blood was harvested from two mice each at days 3, 4, 5, or 6 post-IAV-infected mice and collected in 1.8 mL microcentrifuge tubes (Eppendorf, Framingham, MA, USA). Serum was isolated after blood was allowed to clot at 37 °C for 60 min (m). Clotted blood was centrifuged for 5 m at 12,000× *g* (Minispin, Eppendorf, Framingham, MA, USA), and sera were obtained. Animal study protocols were approved by the Institutional Animal Care and Use Committee (IACUC) at the University of Central Florida (UCF), and euthanasia procedures followed the American Veterinary Medical Association (AVMA) guidelines. 

To pre-treat HBPE-NPs with sera from IAV-infected mice, we followed the protocol optimized in Nierenberg et al. [19] based on the methods presented by Zheng et al. [29]. Briefly, 0.1 mg of PBS-dissolved HBPE-NPs were mixed with 20 μg of serum from 3, 4, 5, or 6 days post-IAV infection in a total volume of 10.5 μL and incubated for 15 m as previously described [19], before use in experiments. These conditions are in the range for optimal absorption of proteins by NPs, about 200 μg of protein per mg NPs. 

### 2.3. Dynamic Light Scattering (DLS) Analysis and IgG Detection

The diameter and surface charge (zeta potential) of HBPE-NPs by DLS were determined as previously described [19] (Zetasizer ZS90, Malvern Panalytical, Malvern, Worcestershire, United Kingdom). To determine whether HBPE-NPs absorbed select immune-related proteins from serum, anti-mouse IgG (Fab-specific) goat antibodies (MilliporeSigma, Burlington, MA, USA) were used to detect absorbed proteins. Subsequent changes in the size of HBPE-NPs upon antibody binding were assessed as in Zheng et al. [30]. Briefly, 0.1 mg of HBPE-NPs was incubated with 20 μg of serum, followed by the addition of 2 μg anti-IgG antibody before DLS analysis as described above. 

### 2.4. Cell Culture

MDA-MB-231 cells (HTB-26), HUVEC cells (CRL-1730), and THP-1 cells (TIB-202) (ATCC, Manassas, VA, USA) were grown in Dulbecco’s Modified Eagle Medium (DMEM) (Corning, Corning, NY, USA), Ham’s F-12K (Kaighn’s) medium (ATCC, Manassas, VA, USA), or Roswell Park Memorial Institute Medium (RPMI)-1640 (ATCC, Manassas, VA, USA) culture media, respectively. Media were supplemented with 10% fetal bovine serum (FBS) (Gemini Bio-Products, Sacramento, CA, USA), penicillin-streptomycin (50,000 units penicillin, 50 mg streptomycin) (Corning, Corning, NY, USA), and 2 millimolar (mM) L-glutamine (Corning, Corning, NY, USA), and cells cultured at 37° Celsius (C) under 5% carbon dioxide (CO_2_). F-12K media were additionally enriched with 56 mg of heparin sodium salt from porcine intestinal mucosa (Corning, Corning, NY, USA) and 15 mg of endothelial cell growth supplement (ECGS) (Corning, Corning, NY, USA) for HUVEC cell growth. RPMI-1640 media were enriched with 50 molar (M) 2-mercaptoethanol (MilliporeSigma, Burlington, MA, USA) for THP-1 monocyte cell proliferation. Green fluorescent protein (GFP)-expressing HUVEC (HUVEC-GFP) cells (Essen BioScience, Ann Arbor, MI, USA) used F-12K media as described above. MDA-MB-231 cells expressing luciferase (MDA-MB-231/Luc) (Cell Biolabs Inc., San Diego, CA, USA) used DMEM media as described above. All cell lines were used at a low passage number.

### 2.5. Viability Assay

To evaluate the toxic effects of HBPE-NP treatment on cells, an MTT (3-(4,5-dimethylthiazolyl-2)-2,5-diphenyl tetrazolium bromide) (MP Biomedicals, Irvine, CA, USA) viability assay was performed. Cells were grown to 60% confluency before treatments. MDA-MB-231, HUVEC, or THP-1 cell lines were seeded at an initial density of 0.5 × 10^4^ cells and treated with 10 μL of deionized water (vehicle control), 0.1 mg of PEG-HBPE-NPs, or 0.1 mg of HBPE-NPs (pre-treated with sera). To determine viability, 50 μg of MTT reagent was added to cells after 24 h (h) post-treatment as previously described [19]. Absorbance was read at 570 nm (Cytation 5 multi-mode reader, BioTek, Winooski, VT, USA).

### 2.6. In Vitro Assays and Imaging

To test drug delivery efficiency using HBPE-NPs, aliquots of 0.5 × 10^4^ MDA-MB-231, HUVEC, or THP-1 cells were dispensed in 96-well culture plates (Corning, Corning, NY, USA) or 24-well glass-bottom culture plates (Cellvis) and cells were grown to 60% confluency. MDA-MB-231 cells were treated with free Taxol (~50 nM/43 μg), or 0.01 mg of HBPE-NPs (pre-treated with sera) or PEG-HBPE-NPs that were loaded with ~0.5–0.6 μg Taxol. To track cell uptake of HBPE-NPs, cells were incubated with 0.1 mg of HBPE-NPs (pre-treated with sera) or PEG-HBPE-NPs that were loaded with ~0.8–10 μg DiI. 

For imaging cells following treatment, media were replaced with PBS and then substituted with 10% neutral buffered formalin. Cells were imaged via a Zeiss 710 Laser Scanning Microscope (LSM) microscope (Carl Zeiss AG, Dublin, CA, USA). MDA-MB-231 and THP-1 cells were imaged at a 20× magnification, while HUVEC cells were imaged at a 10× magnification. Total nanoparticle presence within a cell was determined through the detection of DiI fluorescence by imaging with a Cytation 5 multi-mode reader at a 10× magnification and quantifying using ZEN blue software (Carl Zeiss AG, Dublin, CA, USA). For the quantification of images, all data were representative of three different fields of view. 

### 2.7. Chemotactic Transwell (CT) Protocol

To assess the interaction of HBPE-NPs with endothelial cells, GFP-expressing HUVEC (HUVEC-GFP) cells were dispensed in a ClearView 96-well chemotaxis plate (Essen BioScience, Ann Arbor, MI, USA) as previously described [19]. Plate inserts were coated with 0.75 μg fibronectin (MilliporeSigma, Burlington, MA, USA), and HUVEC-GFPs were seeded at an initial density of 5 × 10^4^ cells per well. Cells were grown to 80% confluency and incubated with 0.1 mg of PEGylated or sera-treated HBPE-NPs as per our published protocol [19]. Bright-field and fluorescent time-lapse images of endothelial cells and HBPE-NPs were acquired using an IncuCyte S3 Live-Cell Analysis System (Essen BioScience, Ann Arbor, MI, USA) at 10× magnification. Cells were imaged at 30 m or 60 m intervals for 24–48 h. Chemotaxis software (Essen BioScience, Ann Arbor, MI, USA) was utilized for the quantification of total fluorescence and overlay of fluorescence (merged signals from cells and HBPE-NPs). Time-lapse videos were generated. The migration of HUVEC-GFP cells from the insert to the reservoir plate was assessed by tracking movement from the top to the bottom of the insert for 48 h, imaging the cells every 30 m. 

### 2.8. Modified Transwell Assay

A modified transwell system was used to measure the uptake of HBPE-NPs by cancer cells after interactions with endothelial cells [19]. Briefly, a Millicell- 24 cell culture insert plate (MilliporeSigma, Burlington, MA, USA) was seeded with 3 × 10^4^ HUVEC cells that were 80% confluent after 48 h. The insert was transferred to a 24-well glass-bottom plate (Cellvis, Mountain View, CA, USA), previously seeded with 5 × 10^4^ MDA-MB-231 cells grown to 60% confluency. To the insert with HUVEC cells, 0.1 mg of PEGylated or sera-treated HBPE-NPs was added. After 24 h, to assess the uptake of HBPE-NPs, MDA-MB-231 cells were fixed for imaging individual cells (LSM 710 confocal microscope) or total fluorescence (Cytation 5 multi-modal plate reader). Confocal images were taken at a 20× (MDA-MB-231, THP-1) or 10× magnification (HUVEC). Cytation images were taken at a 10× magnification. Fluorescence was analyzed with ZEN Blue software (Carl Zeiss AG, Dublin, CA, USA).

### 2.9. Biodistribution Studies

To examine the biodistribution of HBPE-NPs *in vivo*, 8-week-old Fox1-nu/nu (nude) female mice were orthotopically implanted with 8 × 10^5^ MDA-MB-231/Luc cells in the mammary fat pad. When tumor volumes reached ~1000 mm^3^, HBPE-NPs loaded with DiR dye (1 mg) were intravenously injected via the tail vein. After 7 h, mice were euthanized, and organs were harvested for imaging of DiR signal (IVIS Lumina S5 *in vivo* imaging system, PerkinElmer, Waltham, MA, USA). Living Image Software (PerkinElmer, Waltham, MA, USA) was used for image analysis. Animal studies were approved by the IACUC at UCF, and euthanasia procedures followed the AVMA guidelines.

### 2.10. Gel Electrophoresis and Immunoblotting

To assess the proteins absorbed from sera by HBPE-NPs, sodium dodecyl sulfate- polyacrylamide gel electrophoresis (SDS-PAGE) was performed. HBPE-NPs were pre-treated with sera for 15 m under conditions of 0.1 mg nanoparticle: 20 μg sera or 0.1 mg nanoparticle: 80 μg sera as described above. To collect HBPE-NPs, samples were centrifuged at 17,000× *g* for 30 m (Optima TLX ultracentrifuge, Beckman Coulter, Miami, FL, USA). Cell pellets were recovered and washed with phosphate-buffered saline (PBS) (Corning, Corning, NY, USA) followed by resuspension in Orange G loading buffer (BioVision Inc., Milpitas, CA, USA) containing 20 mM 2β-mercaptoethanol. Samples were heated at 90 °C for 5 min and centrifuged at 16,000× *g* for 10 s (5415R centrifuge, Eppendorf, Framingham, MA, USA) to remove particulates. Samples (10 mg) were loaded onto a polyacrylamide gel for analysis as previously described [19]. Gels were fixed (30% water, 60% methanol, and 10% acetic acid) for 1 h, stained with a Coomassie Brilliant Blue G-250 (Thermo Fisher Scientific, Waltham, MA, USA) dye for 1 h, and destained (60% water, 30% methanol (Thermo Fisher Scientific, Waltham, MA, USA), and 10% acetic acid (Thermo Fisher Scientific, Waltham, MA, USA). Gels were then imaged with a ChemiDoc MP imaging system (Bio-Rad Laboratories, Hercules, CA, USA). Protein bands on gels were quantified with ImageJ software (National Institutes of Health, Bethesda, MD, USA). 

To confirm that HBPE-NPs absorbed select proteins from sera, immunoblotting was performed. Sera-absorbed HBPE-NPs were pre-treated with sera as described above, 0.1 mg HBPE-NP: 20 μg serum, and SDS-PAGE performed. Proteins were transferred using a transfer buffer (1.9 M glycine (MilliporeSigma, Burlington, MA, USA), 250 mM Tris(hydroxymethyl)aminomethane) (Tris) (Thermo Fisher Scientific, Waltham, MA, USA), 1 mM ethylenediaminetetraacetic acid (EDTA) (Thermo Fisher Scientific, Waltham, MA, USA) in 20% methanol) to a Immobilon-FL polyvinylidene fluoride (PVDF) transfer membrane following the semi-dry procedure (Trans-Blot SD Semi-Dry transfer cell, Bio-Rad Laboratories, Hercules, CA, USA). To quantitate total protein for loading normalization, the membrane was stained with REVERT 700 total protein stain (LI-COR Biosciences, Lincoln, NE, USA) and imaged at 700 nm (Odyssey Imaging System, LI-COR Biosciences, Lincoln, NE, USA). To detect TSP-1, the membrane was treated with UltraCruz Blocking Reagent blocking buffer (Santa Cruz Biotechnology Inc., Dallas, TX, USA) for 60 m and incubated with 1 µg of anti-TSP-1 Alexa Fluor 790 antibody (A6.1) (Santa Cruz Biotechnology Inc., Dallas, TX, USA) overnight at 4 °C. Membranes were then visualized at 800 nm (Odyssey imaging system, LI-COR Biosciences, Lincoln, NE, USA) and normalized using Image Studio Lite software (LI-COR Biosciences, Lincoln, NE, USA).

### 2.11. Identification of Proteins Absorbed by NPs Using Mass Spectrometry

HBPE-NPs were incubated with VS3 or VS5 for 15 m at 0.1 mg HBPE-NPs: 20 μg sera. Samples were processed as described above to collect pellets containing NPs that were subsequently washed in 2 M of ammonium bicarbonate buffer (MilliporeSigma, Burlington, MA, USA). Uncentrifuged and centrifuged VS3 and VS5 sera-only controls were used for comparison. Samples were submitted to the Interdisciplinary Center for Biotechnology Research (ICBR) Proteomics & Mass Spectrometry Facility at the University of Florida for analysis. 

Briefly, protein samples were digested with trypsin to create peptide fragments, followed by trypsin inhibition. Peptide fragments were analyzed via liquid chromatography tandem mass spectrometry (LC-MS/MS) using an Orbitrap Fusion Tribird (Thermo Fisher Scientific, Waltham, MA, USA) coupled to an EASY-nLC system (Thermo Fisher Scientific, Waltham, MA, USA) liquid chromatographer. Data were acquired using Scaffold 5.0 software (Proteome Software, Portland, OR, USA). Proteins were identified using a Mascot 2.7 search engine (Matrix Science, Boston, MA, USA) and the Universal Protein Resource (UniProt)_Mus_20200805 database. A parent mass error tolerance of 10-ppm was utilized. Fixed modifications for Mascot 2.7 searching were the oxidation of pyrrolysine and carbamidomethylation of cysteine. Variable modifications included N-terminal pyro-glutamate, deamidation of asparagines and glutamine, and oxidation of methionine. Spectral counts of individual proteins were normalized to the average total spectral count among sample sets divided by the total spectral count for a particular sample set.

### 2.12. Statistical Analysis and Databases 

Prism 8 software (GraphPad Software, Inc., San Diego, CA, USA) was used to determine the statistical significance of the data. Welch’s correction of a Student *t*-test was applied to parametric, unpaired, and two-tailed conditions. This allowed statistical assessment between two experimental groups of data. *p*-values < 0.05, representing a 95% confidence level are designated in figure legends and were deemed statistically significant. Determination of shed proteins with immune-related function and their probability of shedding (S-score) were acquired through the A Deep Learning Model for Predicting the Shedding Events of Membrane Proteins (DeepSMP) (http://www.csbg-jlu.info/DeepSMP/) and SheddomeDB databases [31] last accessed on 9 April 2021. Function, ligands, and potential cancer interaction of mass spectrometry-identified proteins in HBPE-NP coronas were determined by accessing the UniProt database (https://www.uniprot.org/) and Gene Expression and Mutations in Cancer Cell Lines (GEMICCL) (https://www.kobic.kr/GEMICCL/) database or by literature review. These databases were last accessed on 18 March 2021 and 2 April 2021, respectively.

## 3. Results

### 3.1. Formation of Protein Corona on HBPE-NPs Using Sera from IAV-Infected Mice

Research from our lab and others previously showed that polymeric NPs treated with serum from healthy mice (e.g., normal sera) absorbed macromolecules that enhanced tumor cell uptake [19,32]. Collectively, these studies suggest that the absorbed proteins that form the protein corona strongly modulate the biological identity of NPs; hence, defining a protein or groups of proteins that most positively influence the cancer cell uptake of NPs could advance the clinical utility of NPs for drug delivery. To this end, we surmised that sera with increased protein content, such as resulting from an immune response to infection, would be a novel source for the discovery of proteins selectively absorbed by NPs that could improve the efficiency of drug delivery to tumors. 

During an infection, proteins are shed from antigen-activated immune cells into the lymph and bloodstream. This is shown in a list of predicted and known shed proteins with immune-related activity and their probability of shedding (S-score, generated using DeepSMP (A Deep Learning Model for Predicting the Shedding Events of Membrane Proteins) [http://www.csbg-jlu.info/DeepSMP/] and SheddomeDB databases [31] (Appendix A). During infections, proteins shed into circulation function in biological processes like leukocyte migration and immune cell activation, stimulating B cells, T cells, and natural killer (NK)-cells. Utilizing serum from infected hosts could thus provide a rich source of coronal proteins for absorption by NPs. 

Our previous work optimized the synthesis of a hyperbranched polyester (HBPE) polymer to form polymeric HBPE-NPs [19]. HBPE-NPs, when first prepared, are carboxylated (COOH) and negatively charged (Table 1). Previously, we showed that COOH-HBPE-NPs (hereafter referred to as HBPE-NPs) that are unmodified (e.g., not pre-treated with sera or PEGylated) would non-specifically absorb macromolecules from fluids and, under *in vivo* conditions, are rapidly cleared from the circulation, limiting their utility [19]. Herein, we used HBPE-NPs pre-coated with sera obtained from mice infected with IAV and loaded with the tracking dye, DiI or DiR, or the cancer drug, Taxol, to investigate the characteristics of the protein corona formed and impact upon the biological behavior of particles. As a non-sera-treated control for these experiments, we used the PEGylated particles (PEG-HBPE-NPs), as these are the standard in the field. As previously described, the equal loading of dye or drug in all particles was confirmed [19]. 

The murine IAV model was chosen as the source for sera to pre-treat particles since infection proceeds from a stealth phase (days 0–2) with minimal weight loss, to activation of innate (days 2–4) and adaptive (days 4–7) immunity, with significant weight loss and physical deterioration post-7 days (Figure 1A) [33,34], enabling evaluation of proteins shed at differences phases of the immune response. We collected sera from 2–3-month-old female IAV-infected C57BL/6 mice that were inoculated intranasally with a 300 LD50 lethal dose of influenza at days 3-, 4-, 5-, and 6-days post-infection, termed VS3, VS4, VS5, and VS6, and used these sera for pre-forming protein coronas on HBPE-NPs. Sera-treated NPs are denoted by the NP abbreviation (HBPE-NPs) followed by the notation VS3, VS4, VS5, or VS6. To form a protein corona on HBPE-NPs, 0.1 mg NPs and 20 μg sera were mixed for 15 m under gentle agitation, as previously described [19]. These conditions are optimal for the absorption of proteins by HBPE-NPs (about 200 μg/mg NPs), as we and others have shown [19,29], and were determined by assessing the minimum sera amount and treatment time that induced a notable change in the diameter of the particles.

To detect the absorption of proteins from VS3-6 by particles, the diameter of the HBPE-NPs was analyzed, after incubation with sera, by dynamic light scattering (DLS) using a modified protocol from Zheng et al. [30]. Briefly, when HBPE-NPs bind sera proteins, such as immunoglobulin G (IgG), anti-IgG-specific antibodies can be used to detect these absorbed proteins, which changes the diameter of the HBPE-NPs detected by DLS [30]. After treatment with VS3-6, HBPE-NPs decreased in average diameter (Table 1), an effect previously noted and likely due to sera proteins constricting the branching of the HBPE polymer [19]. The addition of anti-IgG antibodies led to increases in the average diameter of HBPE-NPs (VS3), HBPE-NPs (VS4), HBPE-NPs (VS5), and HBPE-NPs (VS6) by ~21.9%, ~12.3%, ~6.3%, and ~11%, respectively, indicating that HBPE-NPs absorbed sera proteins (like immunoglobulins) produced by IAV-infected mice. While not conclusive, these data suggest that differences in the size of HBPE-NPs could be attributed to the absorption of unique proteins depending on the sera collection day and the immune response status. This was subsequently confirmed upon identification of sera proteins by mass spectrometry, as will be further discussed. Zeta potential was evaluated to determine if the sera from IAV-infected mice altered the surface charge of HBPE-NPs, which can influence the cellular uptake of NPs [20,21]. The average surface charges of non-sera-treated HBPE-NPs and HBPE-NPs (VS3-6) were negative and indicated stability but on the threshold of tending towards aggregation (Table 1). 

Having shown that HBPE-NPs could absorb macromolecules from VS3-6, studies were carried out to exclude any possible cellular toxicities due to nanomaterials. MDA-MB-231 (breast cancer), HUVEC (endothelial), or THP-1 (monocytic) cells were incubated with DiI-loaded HBPE-NPs, pre-treated with VS3-6, and, after 24 h, an MTT assay was performed to assess changes in viability (Figure 1B–D). The results were that pre-treatment of HBPE-NPs with VS3-6 did not produce particles that induced cell death in treated cells which could be attributed to inherent toxicity. The viability of cells treated with sera-coated HBPE-NPs was equivalent to the vehicle control. We previously showed that untreated HBPE-NPs and PEG-HBPE-NPs were also not toxic [19].

### 3.2. Direct Uptake of HBPE-NPs Pre-Treated with Sera from IAV-Infected Mice by Monocytic, Endothelial, and Breast Cancer Cells

To determine whether pre-treating HBPE-NPs with VS3-6 could modulate cell uptake, MDA-MB-231, HUVEC, and THP-1 cells were incubated with VS3-6-pre-treated HBPE-NPs. The PEG-HBPE-NPs were used as controls. Each formulation of HBPE-NPs before sera treatment or PEGylation was encapsulated with DiI dye for tracking fluorescent particles in cells. All HBPE-NPs had comparable loading of encapsulated DiI; see Nierenberg et al. [19]. Laser confocal scanning microscopy was used to visualize particles within cells, and mid-plane images of cells were acquired. These data are not quantitative and are shown for visualization purposes. For quantitative measurements of particle uptake, a multi-modal plate reader with digital imaging capacity was used to determine total fluorescence within a population of cells. Fluorescence values were calculated by averaging the pixel intensity of individual cells.

Cellular uptake of VS3-6 treated HBPE-NPs, and PEG-HBPE-NPs was visualized 24 h post-treatment and showed that all cell types could internalize particles (Figure 2A). DiI fluorescence, representative of total particle uptake per cell, was quantified only from digital images (Figure 2B–D). In MDA-MB-231 cells, VS3-6-treated HBPE-NPs exhibited a significant (*p* < 0.0001) enhancement in the mean uptake of particles as compared to PEG-HBPE-NPs (Figure 2B). In HUVEC cells, HBPE-NPs (VS5-treated and V6-treated) had a reduced average uptake by ~13.3% (*p* < 0.004) and ~27.2% (*p* < 0.0001), respectively, relative to PEG-HBPE-NPs (Figure 2C). In THP-1 cells, HBPE-NPs (VS3-treated) demonstrated a mean increase in uptake (*p* < 0.02) by ~17.4%, while HBPE-NPs (VS4-6 treated) showed a reduction in uptake (*p* < 0.0001) by ~63.9%, ~54.6%, and ~64.8%, accordingly, compared to PEG-HBPE-NPs (Figure 2D). Hence, HBPE-NPs pre-treated with VS4-6 showed enhanced cell delivery to cancer cells over PEG-HBPE-NPs and reduced uptake by monocytes. The modest reduction in uptake of HBPE-NPs (VS5-6-treated) by HUVEC cells may be attributed to the transient residence of particles due to cellular transport processes, like endocytosis and exocytosis. Hence, HBPE-NPs absorbed select macromolecules from VS3-6 and formed a protein corona that modulated immune cell and cancer cell uptake compared to PEGylated particles.

### 3.3. Drug Delivery to Cancer Cells Using HBPE-NPs Pre-Treated with Sera from IAV-Infected Mice

To assess if uptake of HBPE-NPs (VS3-6) by MDA-MB-231 cells correlated with enhanced delivery of cancer-killing drugs compared to PEGylated particles, HBPE-NPs were encapsulated with ~0.5–0.6 μg Taxol and subsets of particles either pre-treated with VS3-6 or PEGylated. After 24 h post-treatment, cell viability was evaluated using an MTT viability assay (Figure 3). PBS alone was used as vehicle control. Free Taxol (~50 nM/43 μg) was used as a positive control. Equivalent Taxol loading in all HBPE-NPs before sera treatment was determined using methods described in Nierenberg et al. [19]. While VS3-6-treated HBPE-NPs and control PEG-HBPE-NPs delivered toxic doses of Taxol, cancer cells incubated with HBPE-NPs (VS4 or VS5) demonstrated a statistically significant reduction in viability as compared to PEG-HBPE-NPs. Increased drug-mediated toxicity may be attributed to HBPE-NPs (VS4 or VS5) promoting higher NP uptake in MDA-MB-231 cells over PEG-HBPE-NPs (Figure 2B and Appendix A). HBPE-NPs (VS4 or VS5 treated) effectively delivered a dose of Taxol (~0.5 μg) that decreased the viability of breast cancer cells by ~48% and ~54%, respectively, and was ~80-fold lower than the IC50 dose of free Taxol (43 μg) that resulted in an equivalent amount of cell death. These results indicate that by absorbing macromolecules from VS4-5, drug delivery to cancer cells by HBPE-NPs was more efficient.

### 3.4. Cancer Cell Uptake of HBPE-NPs Pre-Treated with Sera from IAV-Infected Mice after Interaction with an Endothelial Layer

To reach tumors, NPs may traffic through endothelial cells that line the blood vessels supplying tumors with oxygen and nutrients; hence, the ability of a drug to cross an endothelial barrier could be improved with nanoparticle-mediated delivery [35,36]. To determine whether the VS3-6-treated HBPE-NPs can cross an endothelial layer and be taken up by cancer cells, we used a modified transwell system that mimics aspects of the cellular interactions observed with *in vivo* delivery conditions. The transwell system features an HUVEC-seeded insert with a porous membrane placed in a plate (bottom) seeded with MDA-MB-231 cells (adopted from Nierenberg et al. [19]). This setup allows the *in vitro* evaluation of sera-treated HBPE-NPs and PEG-HBPE-NPs, as these are taken up by cancer cells after moving through an endothelial layer. Previous work confirmed that the presence of HUVECs is necessary for the transition of HBPE-NPs from the insert to the bottom plate of the transwell system since, in the absence of HUVECs, most HBPE-NPs remain in the insert [19]. We also optimized the confluency of the endothelial layer at 80% and demonstrated that the movement of HBPE-NPs from the insert to the bottom plate did not vary significantly with the confluency of the endothelial cells [19]. Importantly, we also demonstrated that the interaction of HUVECs with VS3-6-treated HBPE-NPs did not stimulate the migration of HUVECs that could potentially promote angiogenesis (Appendix A). 

Using the modified transwell system, we ascertained if VS3-6-treated HBPE-NPs loaded with DiI dye could localize from the insert lined with endothelial cells (HUVECs) to the bottom chamber seeded with breast cancer cells. PEG-HBPE-NPs were used as a standard control. We added 0.1 mg HBPE-NPs (VS3-6-treated) or PEG-HBPE-NPs to the insert with HUVECs and placed the insert in the bottom plate containing MDA-MB-231 cells for 24 h before measuring uptake of HBPE-NPs by MDA-MB-231 cells. The uptake of DiI-loaded sera-treated HBPE-NPs and PEG-HBPE-NPs by cancer cells was visualized by confocal microscopy (Figure 4A) and quantitated by digital microscopy (Figure 4B and Appendix A). Note that the 0.1 mg dose of HBPE-NPs for the transwell experiments was optimized in Nierenberg et al. [19]. Uptake by MDA-MB-231 cells of HBPE-NPs (VS5-treated) and HBPE-NPs (VS6-treated) was significantly higher (*p* < 0.0001) than PEG-HBPE-NPs by ~48.6% and ~44.2%, respectively, indicating improved cancer cell uptake of HBPE-NPs, even after prior exposure to endothelial cells. Recall that in the direct uptake experiments with MDA-MB-231 cells, we found that HBPE-NPs (VS3-6) demonstrated higher uptake than PEG-HBPE-NPs (Figure 2B). This did not change in the two-chamber transwell studies in which HBPE-NPs pre-treated with VS3-6 were initially exposed to HUVECs (Figure 4A,B). Hence, we concluded that the movement of HBPE-NPs, pre-treated with VS3-6, from the insert into the bottom plate to be taken up by cancer cells was not impaired by the initial contact of HBPE-NPs with endothelial cells and is an area for future optimization to improve the transcytosis of systemically introduced HBPE-NPs. 

### 3.5. In Vivo Biodistribution of HBPE-NPs Pre-Treated with VS5

HBPE-NPs pre-treated with VS5 provided the best results in terms of improved uptake by breast cancer cells under all conditions tested (Figure 2B, Figure 3 and Figure 4B) with reduced uptake by monocytes (Figure 2D). Using VS5-treated HBPE-NPs compared to untreated HBPE-NPs, we examined the *in vivo* biodistribution of these particles loaded with DiR dye in tumor-bearing mice. PEG-HBPE-NPs were included as a control. We used an orthotopic model of triple-negative breast cancer (TNBC) in which MDA-MB-231-luc2 cells were orthotopically implanted into the mammary fat pad of nude female mice. Once tumors grew to approximately 1000 mm^3^, DiR infrared dye-loaded HBPE-NPs, untreated or pre-treated with VS5, or dye-loaded PEG-HBPE-NPs were systemically injected into mice through the tail vein. After 7 h, mice were euthanized, and organs were harvested (tumor, heart, lungs, spleen, kidneys, liver) to quantify the fluorescent signal from the HBPE-NPs by imaging. Compared to untreated HBPE-NPs, there was increased tumor uptake of HBPE-NPs (VS5-treated) that was comparable to the standard modification of particles as seen with the PEG-HBPE-NPs (Figure 5A,B). Importantly, decreased spleen and liver uptake of HBPE-NPs pre-treated with VS5 compared to PEGylated particles were detected (Figure 5A,B), which suggests that more of these particles could stay in circulation and accumulate for a longer time in tumors. This is exemplified by the significant decrease in kidney uptake observed for HBPE-NPs (VS5-treated) compared to untreated HBPE-NPs. These results, in total, suggest that the pre-treatment with VS5 modulated the biodistribution of HBPE-NPs since the same core particle (HBPE-NPs) and cargo (DiR) were used as in the controls. Hence, pre-treating with VS5 can confer to HBPE-NPs the capacity for improved circulation and tumor accumulation that is the same if not better than PEGylation, with the potential for fewer side effects typically associated with PEGylation. 

### 3.6. Identification of Proteins Absorbed by HBPE-NPs Pre-Treated with VS3 and VS5

We first analyzed the proteins absorbed by HBPE-NPs after pre-treatment with VS3-6 by gel electrophoresis. Two NP:sera ratios were examined: the particle to sera conditions of 0.1 mg HBPE-NPs:20 μg sera used in all experiments and a higher concentration of sera of 0.1 mg HBPE-NPs:80 μg sera to increase protein content for analysis. Proteins absorbed by NPs were isolated by high-speed centrifugation, analyzed by SDS-PAGE, and visualized on gels by Coomassie Blue staining. For the 0.1 mg HBPE-NPs: 80 μg sera condition, the most abundant coronal proteins had molecular weights between 20–25 kD and 50–75 kD, likely corresponding to major blood proteins like albumin and immunoglobulin (~60 kDa) (Appendix A). Differences in the protein patterns in each lane were noted between HBPE-NPs treated with VS3-6 but were most noticeably in the top 30% of the high molecular weight proteins (Appendix A). 

To identify specific proteins absorbed by HBPE-NPs from VS3 and VS5, we used a mass spectrometry approach to analyze the proteome from VS3-treated and VS5-treated HBPE-NPs. Both these sera conferred improved cancer cell uptake but differed in the uptake by monocytes (Figure 2). Protein samples from VS3 and VS5-treated HBPE-NPs were prepared, and controls included VS3 and VS5 alone. We noted several differences between the proteins absorbed by the HBPE-NPs treated with VS3 vs. VS5. HBPE-NPs treated with VS3 absorbed more innate immunity-related proteins, such as complement and acute phase proteins, that could mediate the uptake of particles by monocyte/macrophages. HBPE-NPs treated with VS5 had more proteins involved in coagulation, metabolism, and immunoglobulin formation (Table 2). The functions of identified coronal proteins were assessed via the UniProt database. These results may partly explain the differences in particle uptake by THP-1 cells (Figure 2C) and particle accumulation in the liver/spleen of mice (Figure 5). A complete list of proteins detected in VS3- and VS5-treated HBPE-NPs is shown in Appendix A. 

We further observed that while identified proteins absorbed by HBPE-NPs pre-treated with VS3 and VS5 could interact with cancer cell receptors, proteins absorbed by HBPE-NPs treated with VS5 included those with multi-receptor binding properties. In contrast, proteins absorbed by HBPE-NPs treated with VS3 involved more single-receptor/ligand interactions (Table 3). Potential cancer interactions were determined by searching the GEMICCL database or reviewing the literature. As shown in Table 4, HBPE-NPs treated with VS5 were enriched for common sera proteins such as albumin, pregnancy zone protein, and apolipoproteins found on other types of NPs [37,38,39]. Sera proteins more specific to the HBPE-NPs included apolipoprotein A-I, complement factor I, ceruloplasmin, and TSP-1. While multiple cell types can produce these proteins, many of these proteins are also produced by immune cells, such as T lymphocytes [40,41]. As shown in Table 4, most of these proteins can directly or indirectly promote interactions with MDA-MB-231 cells, suggesting that the proteins absorbed from VS5 may endow the HBPE-NPs with the ability to interact better and be taken up by cancer cells. TSP-1 was of interest for future study since it is rarely found in the protein corona of other polymeric NPs [42,43,44] and can interact with multiple cancer receptors.

To confirm the identification of proteins absorbed from VS5 by HBPE-NPs, we isolated proteins from HBPE-NPs and performed an immunoblot for one of the proteins identified, TSP-1. We observed that TSP-1 was enriched on the HBPE-NPs compared to sera alone (Figure 6A,B and Appendix A), validating the results from the mass spectrometry data (Table 4) and indicating that HBPE-NPs may selectively absorb proteins from sera. Note that equivalent protein amounts were loaded in each sample. 

## 4. Discussion

Studies of the protein corona formed on NPs have often utilized normal sera or plasma from mice or humans as sources of macromolecules [70,71,72,73]. Media containing bovine serum albumin (BSA) and FBS are examples of sera commonly used in studies of the nanoparticle protein corona [74]. The most common coronal protein from these sources is albumin which comprises ~60% of the blood’s protein content [16,75]. Due to its biological prevalence, biocompatibility, and capacity to target tumor receptors, the use of albumin to improve the delivery of cancer drugs or nanomedicines has been extensively explored [76]. However, albumin-based particles can also exhibit low circulation efficiency *in vivo* due to rapid immune clearance [77,78] and induce severe allergic reactions and other harmful side effects in patients [79], possibly attributed to poor tissue-specific targeting, limiting the approach of using such single components to pre-form a protein corona.

Using the capacity of NPs to bind or absorb select macromolecules from complex biofluids could result in a protein corona that confers a unique biological identity to the particles. For example, it was demonstrated that phosphocholine-derived liposomes could passively enrich themselves with shed complement C3 that consequently promoted their localization to pneumonic mouse lungs through neutrophil trafficking [80]. NPs can also form distinct biological identities based on their initial exposure to a protein source. When polystyrene NPs were incubated with either human plasma or human cerebrospinal fluid (CSF), coronas comprising different concentrations of the same protein, or unique proteins, formed [81]. For instance, kininogen-1 was the 5th most abundant coronal protein in plasma-incubated NPs, while it was the 15th most abundant in CSF-incubated NPs. When analyzing the top 20 most abundant coronal proteins, serum albumin was found only in the plasma-treated NP coronas but not CSF-treated. The resulting coronal fingerprints on NPs may be influenced by variations in the concentration of individual proteins or protein classes found in distinct biological fluids. Analysis of the protein coronas from NPs treated with plasma from sepsis-affected and non-affected patients showed that the coronal protein concentrations could differ by factors of 2- to 317-times [82]. The most significant differences involved immune-related proteins, such as immunoglobulins and complement. We similarly showed that HBPE-NPs could absorb different protein content from sera collected during the different stages of an IAV infection, with differences detected between sera collected in the early stage of the immune response to IAV compared to the later stages. This could be due to a dynamic relationship among protein levels in response to infection, which was observed in bronchoalveolar lavages from mice infected with IAV at 5-, 14-, and 21-days post-infection [83]. As an example, haptoglobin levels increased 27-, 4-, and 9-fold on days 5, 14, and 21 post-infection compared to pre-infection levels [83]. Hence, the protein coronas on HBPE-NPs derived from VS3-VS6 likely reflect the variety of macromolecules shed during infection.

Harnessing the unique macromolecular environment present during infection may provide insight into new disease-targeting macromolecules. In one study, researchers explored the effect of incubating NPs with disease-derived biofluids on the delivery of cancer therapies. Chitosan NPs were incubated in FBS, normal human sera, or sera from breast cancer patients to evaluate the *in vitro* uptake of folic acid (FA)-conjugated NPs by MCF-7 breast cancer cells [84]. Breast cancer-derived sera (BrCr) were employed to better represent *in vivo* patient treatment conditions. The uptake of BrCr-coated NPs conjugated with a high concentration of FA by MCF-7 cells was increased compared to FBS- or normal sera-treated NPs. When NPs without FA or low-density FA NPs were compared, cell uptake in all sera-coated NP formulations was similar. When corona content was analyzed among the NPs, it was found that BrCr-coated NPs had a higher abundance of serotransferrin and apolipoproteins. Authors hypothesized that these proteins on BrCr-coated NPs interacted with transferrin and HDL receptors known to be overexpressed in breast cancer. Transferrin targeting was also evaluated in lung cancer. Polystyrene NPs were conjugated with transferrin and pre-coated with sera derived from non-small cell lung cancer (NSCLC) or diabetic and NSCLC co-morbidity patients [24]. NSCLC-derived sera were chosen due to the author’s previous finding that NSCLC sera coating diminished NP uptake in A549 lung adenocarcinoma cells compared to normal patient sera and to resemble clinical treatment conditions. In their latest study, the researchers observed that NPs pre-coated with co-morbid sera were taken up by A549 cells more efficiently than NSCLC sera-coated NPs. When the corona content was analyzed, the co-morbid sera corona had more clusterin, fibrinogen, and acute phase-related proteins than the corona from NSCLC sera. These studies, like ours, demonstrate that a diverse coronal content that is stable on NPs may be most effective in promoting cancer cell uptake. 

In the HBPE-NPs pre-treated with VS3 and VS5, we identified proteins commonly found in other nanoformulations as well as proteins more unique to the HBPE polymer. Common proteins found on most nanoformulations include albumin, complement C3, hemopexin, plasminogen, alpha-2-HS-glycoprotein, and apolipoprotein B-100. Proteins coronas of polymeric NPs, such as those made with poly(lactic-co-glycolic acid) (PLGA) or polycaprolactone (PCL) and absorbed with human sera, included common proteins such as albumin and alpha-2-macroglobulin [85], but also proteins unique to PCL-based NPs such as apolipoprotein A-I, cystatin-A, and complement C5. Coronal proteins mainly associated with PLGA-based NPs also included immunoglobulin kappa variable 3–20. Other studies showed that polystyrene NPs incubated in FBS formed protein coronas containing distinctive proteins like fetuin-B, beta-lactoglobulin, and angiotensinogen among the top 20 most abundant proteins [86]. Interestingly, the protein coronas formed on HBPE-NPs pre-treated with VS3 and VS5 contained some proteins rarely found in other polymeric NPs. HBPE-NPs (VS3-treated) were selectively enriched with inter alpha-trypsin inhibitor heavy chain 4, flavin reductase, and carboxypeptidase N, which interact with hyaluronan, riboflavin, and kinin, respectively (UniProt). Receptors for hyaluronan, riboflavin, and kinin are found on cancer membranes (GEMICCL). HBPE-NPs (VS5-treated) were selectively enriched with ceruloplasmin, TSP-1, serum paraoxonase, mannose C-binding protein, and insulin-like growth factor-binding protein complex, which can bind copper/iron, integrins/EGFR/VEGF-A, high-density lipoprotein (HDL), mannose/fucose, and Insulin-like growth factor 1 (IGF-1) (UniProt). Receptors for these macromolecules are often expressed on cancer cells (GEMICCL). Treating HBPE-NPs with a sera source containing shed proteins from an IAV infection increased the likelihood of enriching the protein corona on NPs with factors that interact with receptors highly expressed on cancer cell membranes (e.g., VS5) or, conversely, factors that enhance immune cell interactions (e.g., VS3).

A unique aspect of our findings was the differences in immune-related components of the protein corona found in NPs treated with VS3 compared to VS4-6. VS3 was collected from the earlier innate immune response to IAV, while VS4-6 was from the transition to the adaptive immune response. The increased uptake of HBPE-NPs (VS3-treated) by THP-1 cells compared to HBPE-NPs (VS5) could be in part explained by HBPE-NP (VS3-treated) coronas being enriched with more monocyte/macrophage-binding proteins compared to HBPE-NPs (VS5). More monocyte/macrophage binding proteins, such as complement components, absorbed by HBPE-NPs (VS3) may increase the overall affinity of THP-1 cells for these particles. Alpha-1-antitrypsin was also more prevalent in the protein coronas of HBPE-NPs (VS3). Given that alpha-1-antitrypsin has a variety of binding partners that can interact with monocytes, including cytokines, heme, and lipoproteins [87,88,89], this protein may influence the corona: THP-1 interface. Differences in the albumin concentration of the protein corona on HBPE-NPs may also contribute to the differential uptake of VS3- and VS5-treated HBPE-NPs by THP-1 cells. Albumin is known to be a carrier protein for many small molecules, such as fatty acids and drugs [90,91], and can bind and sequester cytokines like IL-8 [92]. Albumin on NPs could have dysopsonin-like properties that reduce affinity for cell surface molecules. For example, pre-treating organosilica NPs with BSA reduced the *in vitro* uptake of NPs by J774A.1 and primary peritoneal macrophage (pMAC) cells compared to NPs not pre-treated with BSA [93]. How albumin contributes explicitly to the interactions between HBPE-NPs and THP-1 cells or monocytes, especially in the presence of other serum proteins, requires further study but could involve a continuum of activities such as sequestering molecules that modulate the immune response and altering the interface of the NPs to reduce uptake by phagocytes [94]. 

When comparing the analysis of proteins in the coronas of HBPE-NPs (VS3-treated and VS5-treated), we identified proteins that could bind to single or multiple ligand/receptor classes. The enhanced cancer cell uptake of HBPE-NPs (VS5) might partly result from absorbing proteins with multiple cancer receptor-binding properties. Examples of abundant multi-cancer-target proteins absorbed by HBPE-NPs (VS5) included pregnancy zone protein, ceruloplasmin, and TSP-1. Ceruloplasmin’s affinity for copper and iron may allow it to directly bind MDA-MB-231 cells through their copper and iron transporters, such as copper transporter 1 (Ctr1) and ferroportin, respectively [58,60]. Ceruloplasmin could indirectly interact with cancer cells through a shared affinity with albumin for myeloperoxidase, followed by albumin binding FcRn receptors on cancer cells [55,56,57]. TSP-1 is known to bind integrins, Low Density Lipoprotein Receptor Related Protein-1 (LRP1), EGFR, TGF-beta, urokinase-type Plasminogen Activator (uPA), and VEGF-A, all of which are expressed on, or bind to, cancer cells (GEMICCL). To assess potential HBPE-NP coronal components that enhance cancer interaction, TSP-1 was chosen as an example of a multi-target protein for further validation of its presence in the protein corona of HBPE-NPs (VS5). While only a few studies found that protein coronas of polymeric NPs were enriched with TSP-1 [42,43,44], in one report, TSP-1 was detected in the protein corona of nanogels that featured a more hydrophobic surface [44]. In our study, immunoblot analysis confirmed that TSP-1 was present on HBPE-NPs (VS5). 

In recent years, research on the protein corona of NPs has focused on reducing the toxicity of NPs with a pre-existing protein layer [95], avoiding the immune system through surface modifications of NPs [96], and exploring ways to concentrate key proteins within the corona that may favor association with particular cell/tissue types, such as using a peptide to enrich coronas with apolipoproteins [97]. Our work addresses these same aims by using the unique biophysical properties of HBPE to selectively absorb and enrich proteins from IAV infection-derived sera and form a non-toxic protein corona that enhances the interaction of NPs with multiple cancer cell receptors while reducing the uptake by monocytes. Coronal proteins identified herein from VS5 could be adapted for functionalizing the surface of NPs or conjugated to cancer drugs to directly target tumors or indirectly to target cancer-binding proteins. For instance, physiochemical modifications to NPs for integrin- and hyaluronan-binding are a focus of study for cancer targeting nanomaterials [98,99]. Our use of sera from the IAV infection as a novel source of coronal proteins to modulate the biological identity of HBPE-NPs shows that the drug delivery efficiency of NPs can be optimized by taking advantage of the inherent properties of an activated immune system (source of shed proteins) and the biophysical properties of nanomaterials that control the absorption of proteins encountered during circulation.

## 5. Conclusions

The pre-treatment of HBPE-NPs with sera from mice infected with IAV modulated the cellular uptake of nanomaterials. IAV infection-derived sera may contain more macromolecules than normal sera and form a complex protein corona that promotes NPs–cell interactions due to the increased number of proteins shed during immune cell activation. The NPs–cell interface was modulated using sera from different days of the IAV infection; sera from early in the infection favored uptake of NPs by immune cells, while sera from later in the infection improved cancer cell uptake, drug delivery, and *in vivo* biodistribution to tumors. The protein corona formed on NPs was stable even after transit through endothelial cells. HBPE-NPs, pre-treated with VS5, accumulated in tumors as efficiently as PEG-HBPE-NPs, and fewer particles were taken up by the liver or spleen, which could extend particle circulation. This biodistribution could partly be due to VS5-treated HBPE-NPs containing fewer coronal proteins that bind monocytes/macrophages and increased coronal proteins with affinity for multi-cancer receptors. Examples of these included TSP-1, ceruloplasmin, and pregnancy zone protein. Small changes in tumor particle uptake could have significant consequences upon drug delivery, as shown with VS5-treated HBPE-NPs that could kill tumor cells with 80-fold less Taxol than the free drug. Thus, discovering an optimized protein corona coating for NPs that has the potential to improve drug delivery to cancers could result in substantial changes in treatment efficacy and patient outcomes.

## 6. Patents

Application No.:PCT/US2022/016301. Title: CORONAL PROTEIN-COATED NANOPARTICLES AND USES THEREOF. Filing Date: 14 February 2022.

## Figures and Tables

**Figure 1 biomimetics-07-00219-f001:**
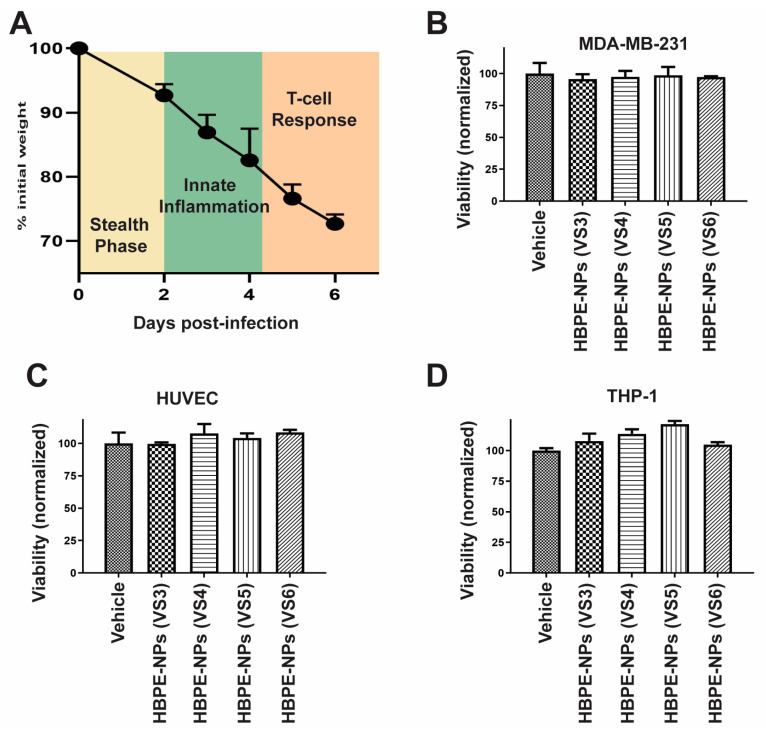
HBPE-NPs pre-treated with sera collected from IAV-infected mice are non-toxic. (**A**) Scheme showing the process of IAV infection in C57BL/6 mice, correlating the phase of the immune response with weight loss. (**B**–**D**) MTT viability assay was performed to assess the toxicity of sera-coated NPs. MDA-MB-231 cells (**B**), HUVECs (**C**), and THP-1 cells (**D**) were treated with vehicle (water), HBPE-NPs pre-coated with sera collected from day 3 post-IAV infection [HBPE-NPs (VS3)], day 4 post-IAV infection [HBPE-NPs (VS4)], day 5 post-IAV infection [HBPE-NPs (VS5)], or day 6 post-IAV infection [HBPE-NPs (VS6)]. Data represent mean ± standard deviation (n = 3).

**Figure 2 biomimetics-07-00219-f002:**
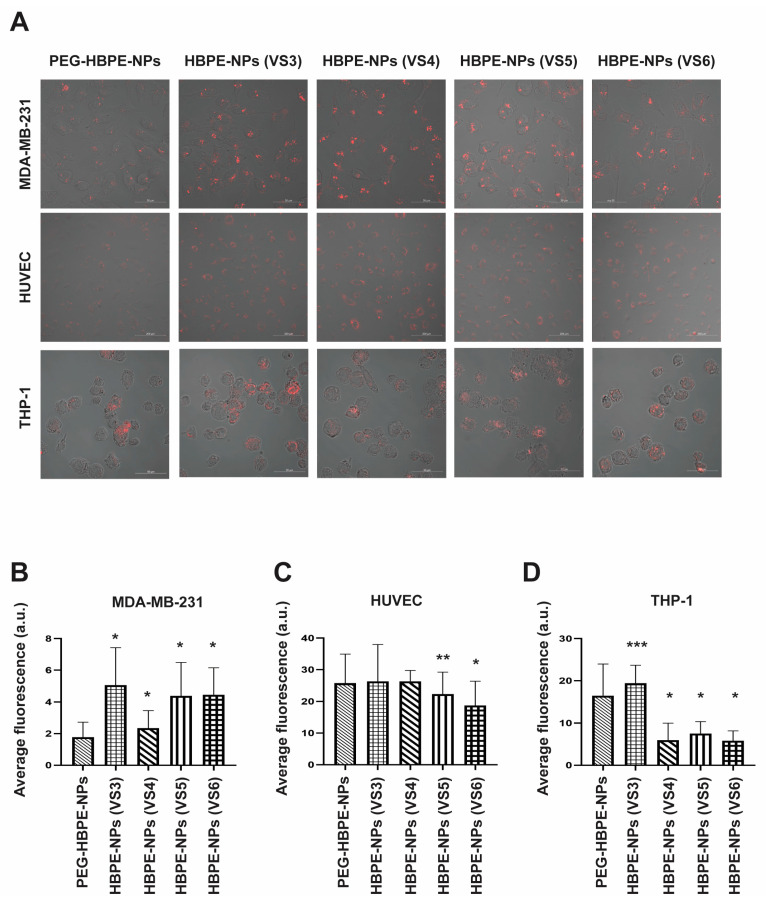
Increased cancer cell uptake and decreased monocyte uptake were observed with HBPE-NPs pre-treated with sera collected from mice at the later stages of the IAV infection. (**A**) HBPE-NPs (NPs) pre-coated with VS3-6 were compared to PEG-HBPE-NPs. Representative confocal microscopic images (upper plane) are shown of MDA-MB-231 cells (upper row), HUVEC cells (middle row), and THP-1 cells (bottom row). In columns are cells treated with DiI dye-encapsulated PEG-HBPE-NPs (first column), HBPE-NPs (VS3) (second column), HBPE-NPs (VS4) (third column), HBPE-NPs (VS5) (fourth column), and HBPE-NPs (VS6) (fifth column). Red fluorescence (DiI dye) is indicative of nanoparticle uptake in cells. The scale bar represents 50, 200, and 50 μm for MDA-MB-231 cells, HUVEC cells, and THP-1 cells, respectively. Images were acquired with a Zeiss LSM 710 microscope at 20× (MDA-MB-231 and THP-1 cells) and 10× (HUVECs) magnifications. (**B**–**D**) Red fluorescence (DiI dye) quantification bar graphs of (**B**) MDA-MB-231, (**C**) HUVEC, and (**D**) THP-1 cells treated with PEG-HBPE-NPs, HBPE-NPs (VS3), HBPE-NPs (VS4), HBPE-NPs (VS5) or HBPE-NPs (VS6) after 24 h. Bar graphs represent the average DiI fluorescence per cell. Quantification data originate from images taken by a Cytation 5 Cell Imaging Multi-Mode Reader (representative images in Appendix A) to capture total nanoparticle fluorescence. MDA-MB-231 and HUVEC fluorescence data were acquired from a minimum of 100 cells. THP-1 fluorescence data were acquired from a minimum of 50 cells. Fluorescence was quantified using Zen Blue software. Data represent mean ± standard deviation. * *p*-value <0.001, ** *p*-value = 0.0034 and *** *p*-value = 0.0199 relative to PEG-HBPE-NPs. Representative data from three experimental replicates are shown.

**Figure 3 biomimetics-07-00219-f003:**
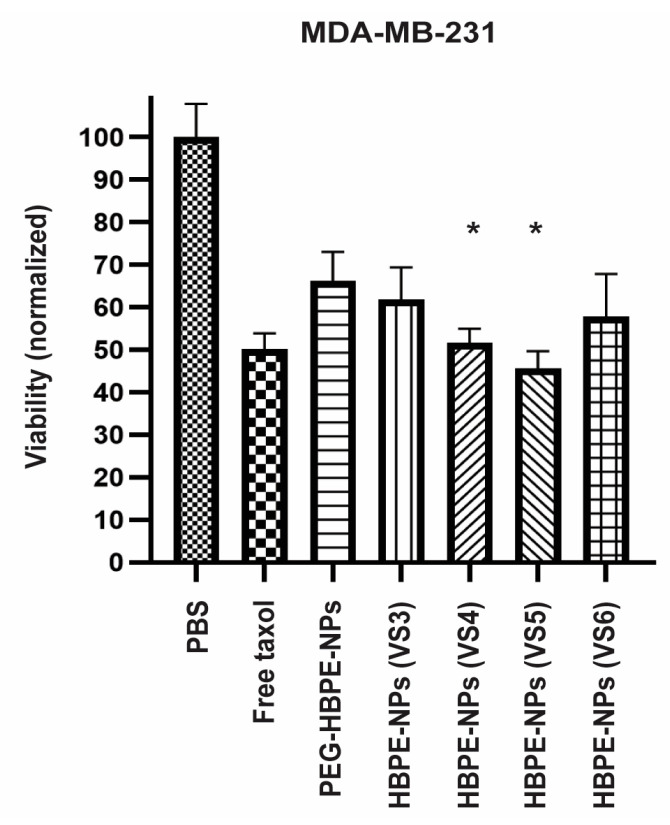
Increased Taxol-mediated toxicity of breast cancer cells when the drug is delivered using HBPE-NPs coated with sera collected from IAV-infected mice. MTT viability assay was carried out to evaluate the toxicity of Taxol-loaded HBPE-NPs. MDA-MB-231 cells were treated with PBS vehicle, free Taxol, PEG-HBPE-NPs, HBPE-NPs (VS3), HBPE-NPs (VS4), HBPE-NPs (VS5), or HBPE-NPs (VS6). Data represent mean ± standard deviation (n = 3). * *p*-value < 0.05 relative to PEG-HBPE-NPs.

**Figure 4 biomimetics-07-00219-f004:**
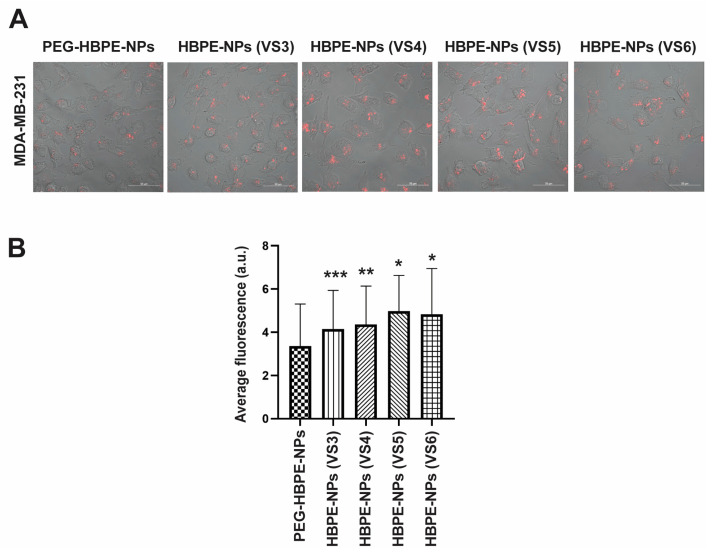
HBPE-NPs pre-treated with sera from IAV-infected mice are taken up by breast cancer cells after interaction with an endothelial layer. (**A**) Transwell experiment in which HUVECs (top chamber) were treated with DiI-loaded HBPE-NPs, and uptake was assessed by MDA-MB-231 cells (bottom chamber). Representative confocal microscopic images of MDA-MB-231 cells are shown after 24 h treatment of HUVECs with PEG-HBPE-NPs, HBPE-NPs (VS3), HBPE-NPs (VS4), HBPE-NPs (VS5), and HBPE-NPs (VS6). The scale bar represents 50 μm. Images were acquired at 20× magnification. (**B**) Red fluorescence (DiI nanoparticle presence) quantification bar graphs of bottom chamber MDA-MB-231cells after 24 h post-treatment of top chamber HUVECs with PEG-HBPE-NPs, HBPE-NPs (VS3), HBPE-NPs (VS4), HBPE-NPs (VS5), or HBPE-NPs (VS6). Average DiI fluorescence per cell was used for quantification from images of 100 cells taken by a Cytation 5 Cell Imaging Multi-Mode Reader (see Appendix A for representative images). Fluorescence was quantified using Zen blue software. Data represent mean ± standard deviation. * *p*-value < 0.0001, ** *p*-value = 0.0002, and *** *p*-value = 0.0028 relative to PEG-NPs. Representative experiments from three replicates are shown.

**Figure 5 biomimetics-07-00219-f005:**
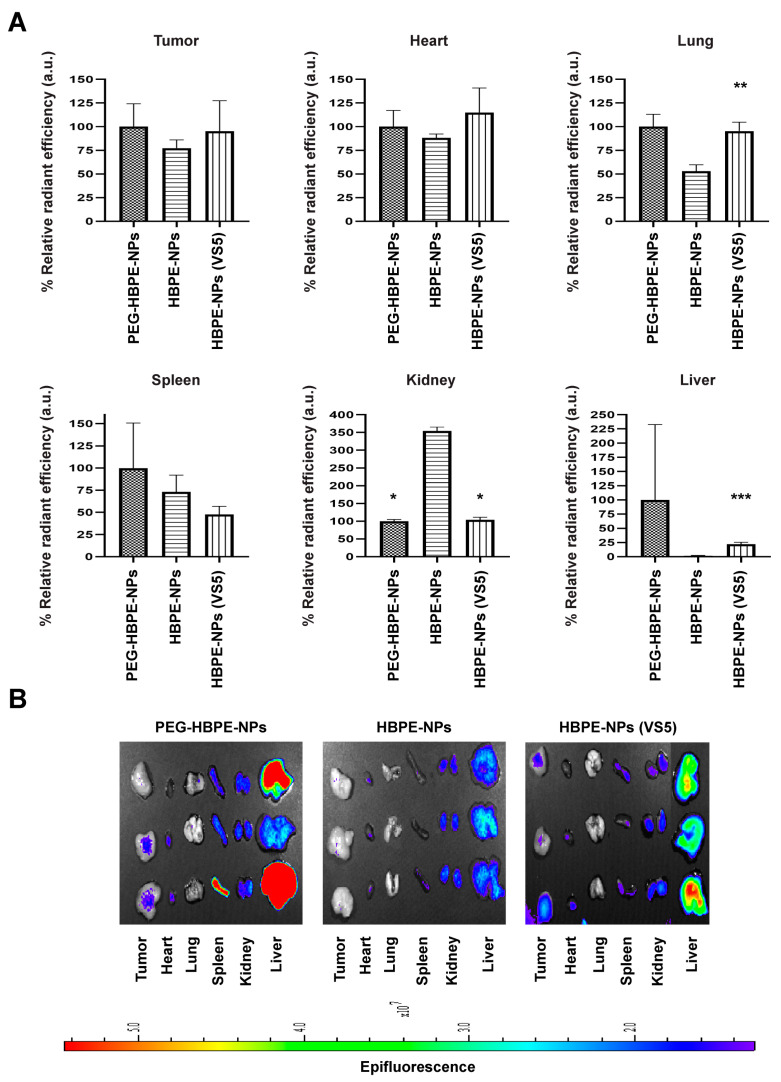
Biodistribution of HBPE-NPs in mice is modulated by pre-treatment with VS5. Nu/Nu nude mice were orthotopically injected with 8 × 10^5^ MDA-MB-231 cells in the mammary fat pad. Upon tumors reaching 1000 mm^3^, mice were intravenously injected with DiR dye-encapsulated PEG-HBPE-NPs, HBPE-NPs, and HBPE-NPs (VS5) and imaged 7 h post-treatment. Images were acquired with an IVIS Lumina S5 and quantified with Living Image software. Fluorescence concentrated in each organ (nanoparticle presence) was quantified relative to PEG-HBPE-NPs; (**A**) bar graphs (upper panels) and (**B**) organ images (lower panels). Images represent nanoparticle uptake (DiR fluorescence) in the tumor, heart, lungs, spleen, kidneys, and liver. Blue color constitutes low nanoparticle accumulation and red/yellow color constitutes high nanoparticle accumulation. Data represent mean ± standard deviation (n = 3). * *p*-value < 0.05, ** *p*-value = 0.0034 and *** *p*-value = 0.0199 relative to HBPE-NPs.

**Figure 6 biomimetics-07-00219-f006:**
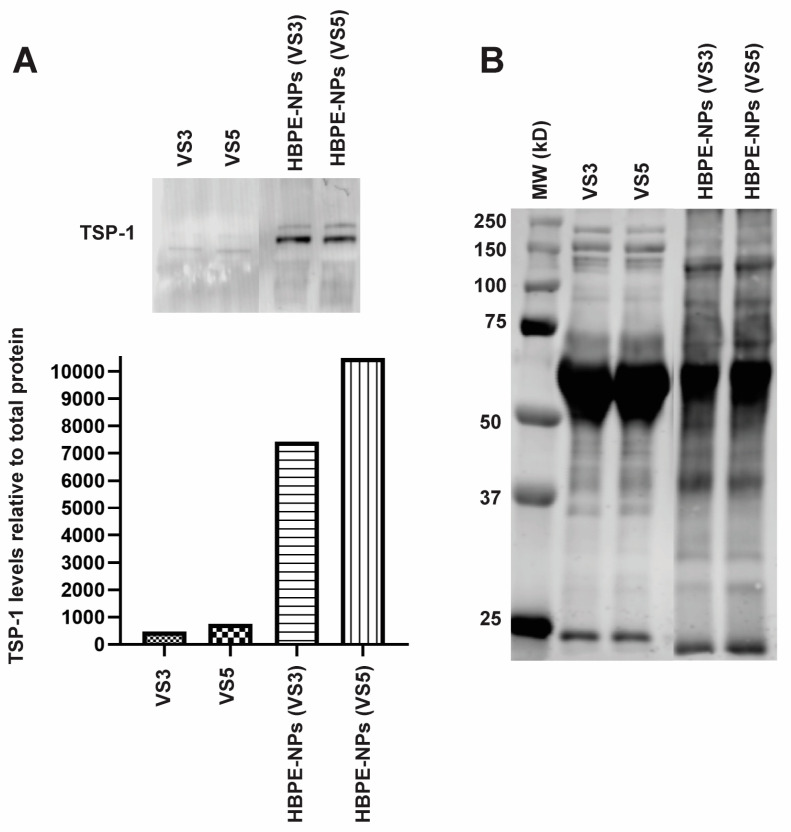
TSP-1 is found in the protein corona formed on HBPE-NPs after treatment with VS5. Proteins absorbed by HBPE-NPs treated with VS3 and VS5 were assessed by SDS-PAGE gels. (**A**) TSP-1 presence in sera alone or on NPs was determined using an anti-TSP-1 antibody (upper panel). The antibody signal was normalized to total protein and quantified using LI-COR Image Studio Lite software (graph, lower panel). (**B**) Total protein per lane was evaluated using REVERT total protein staining. HBPE-NPs were treated with sera at a ratio of 0.1 mg nanoparticle: 20 μg sera. A Precision-Plus Protein Dual Color protein reference ladder was used for molecular weight (MW) comparison. Data shown are representative of two experiments. Raw data is shown in Appendix A. Abbreviations: TSP-1 (thrombospondin-1).

**Table 1 biomimetics-07-00219-t001:** DLS data for HBPE-NPs treated with sera were collected from mice infected with IAV.

Nanoparticle	Particle Size (nm ± σ)	PDI ± σ	ζ-Potential (mV ± σ)
HBPE-NPs	159.0 ± 2.7	0.147 ± 0.040	−32.6 ± 3.8
HBPE-NPs + VS3	145.9 ± 5.2	0.154 ± 0.060	−24.0 ± 0.9
HBPE-NP + VS3 + anti-IgG	177.9 ± 5.7	0.102 ± 0.011	ND
HBPE-NPs + VS4	148.4 ± 4.4	0.183 ± 0.026	−32.8 ± 1.4
HBPE-NPs + VS4 + anti-IgG	166.6 ± 3.9	0.190 ± 0.012	ND
HBPE-NPs + VS5	147.9 ± 6.8	0.179 ± 0.028	−28.0 ± 2.2
HBPE-NPs + VS5 + anti-IgG	157.3 ± 2.3	0.183 ± 0.041	ND
HBPE-NPs + VS6	143.7 ± 2.0	0.174 ± 0.036	−30.4 ± 1.3
HBPE-NPs + VS6 + anti-IgG	159.5 ± 5.1	0.157 ± 0.017	ND

Values represent an average of three replicates. Abbreviations: VS, viral sera collected from IAV-infected mice, numbers refer to the day of sera collection post-infection; PDI, polydispersity index, anti-IgG, anti-IgG antibody, ND, not determined.

**Table 2 biomimetics-07-00219-t002:** Comparison of proteins absorbed by HBPE-NPs from VS3 or VS5 that could promote interactions with monocytes/macrophages, based on known functional data (https://www.uniprot.org/). Abundance was determined by multiplying the individual protein spectral count by (the average total spectral count among samples divided by the total spectral count for an individual sample).

Protein Abundance (VS3 > VS5)	Protein Abundance (VS5 > VS3)
Complement C3	Albumin
Alpha-2-HS-glycoprotein	Fibronectin
Complement factor B	Apolipoprotein B-100
Vitronectin	Complement factor H
Clusterin	Haptoglobin
Inhibitor of carbonic anhydrase	Immunoglobulin heavy constant mu
H-2 class I histocompatibility antigen, Q10 alpha chain	Complement component C8 beta chain
Complement C5	Ig gamma-2B chain C region
Carboxypeptidase N subunit 2	Protein AMBP
Plasma protease C1 inhibitor	Ig gamma-1 chain C region, membrane-bound form
Alpha-1-acid glycoprotein 1	Complement component C8 gamma chain
Alpha-2-antiplasmin	Alpha-1-acid glycoprotein 2
Complement component C8 alpha chain	Immunoglobulin kappa constant
Complement component C9	Mannose-binding protein C
Serum amyloid A-1 protein	Beta-2-microglobulin
Complement factor D	Serum amyloid P-component
Serum amyloid A-2 protein	Complement C1s-B subcomponent
Ig-like domain-containing protein	Transthyretin
Complement C1s-A subcomponent	
N-acetylmuramoyl-L-alanine amidase	
Carboxypeptidase N catalytic chain	
Complement C2	
Complement component 7	
Mannan-binding lectin serine protease 2	
Ficolin-1	
Complement C1r-A subcomponent	
Vitamin K-dependent protein S	
Mannan-binding lectin serine protease 1	
Glyceraldehyde-3-phosphate dehydrogenase	
Vitamin K-dependent protein C	
Interleukin-1 receptor accessory protein	

**Table 3 biomimetics-07-00219-t003:** List of top 10 proteins absorbed by HBPE-NPs treated with VS3 or VS5 with potential cancer interaction (multi-ligand receptors in *italics*) based on data from known ligands, MDA-MB-231 cells (https://www.uniprot.org/, https://www.kobic.kr/GEMICCL/), or reviewing the literature. Abundance was determined by multiplying the individual protein spectral count by (the average total spectral count among samples divided by the total spectral count for an individual sample).

Protein Abundance (VS3 > VS5)		Protein Abundance (VS5 > VS3)	
Protein	Potential Cancer Interaction	Protein	Potential Cancer Interaction
Inter alpha-trypsin inhibitor, heavy chain 4	Hyaluronan	Albumin	*SPARC, hnRNPs, calreticulin, FcRn,* *Cubilin* [45,46,47,48]
Alpha-2-HS-glycoprotein	Transforming growth factor-beta [49]	Pregnancy zone protein	*LRP1, IL-1, GRP78* [50,51,52,53]
Inter-alpha-trypsin inhibitor heavy chain H2	Hyaluronan	Apolipoprotein B-100	*Lipids, lipid receptors*
Clusterin	Low-density lipoprotein receptor	Beta-2-glycoprotein 1	*Phospholipids*
Histidine-rich glycoprotein	*Phospholipids* [54]	Ceruloplasmin	*Albumin (FcRn), Ctr1, ferritin, ferroportin* [55,56,57,58,59,60]
Afamin	*Fatty acids, vitamin E receptors*	Serum paraoxonase/arylesterase 1	High-density lipoprotein
Carboxypeptidase N subunit 2	Kinin receptors	Glutathione peroxidase 3	*Selenium* *(ApoER2 + LRP1 + LRP2)* [61,62,63]
Apolipoprotein A-II	*Lipids, lipid receptors*	Insulin-like growth factor-binding protein complex acid labile subunit	Insulin-like growth factor 1 receptors
Corticosteroid-binding globulin	Glucocorticoid receptors	Beta-2-glycoprotein 1	*Phospholipids*
Flavin reductase (NADPH)	Riboflavin receptors	Apolipoprotein C-III	*Lipids, lipid receptors*

Abbreviations: Reduced nicotinamide adenine dinucleotide phosphate (NADPH), Secreted protein acidic and rich in cysteine (SPARC), Heterogeneous nuclear ribonucleoproteins (hnRNPs), neonatal fragment crystallizable receptor (FcRn), apolipoprotein receptor 2 (ApoER2), low density lipoprotein receptor-related protein 1 (LRP1), low density lipoprotein receptor-related protein 2 (LRP2).

**Table 4 biomimetics-07-00219-t004:** List of most abundant proteins absorbed by HBPE-NPs treated with VS5 that had potential cancer cell interactions based on data from known ligands, MDA-MB-231 cells (https://www.uniprot.org/, https://www.kobic.kr/GEMICCL/), or reviewing the literature. Abundance was determined from spectral counts normalized to the total number of spectra in sera samples (Normalized Total Spectra).

Identified Protein	Accession #	Molecular Weight (kD)	Potential Cancer Interaction (MDA-MB-231)
Albumin	ALBU	69	SPARC, hnRNPs, calreticulin, FcRn, Cubilin [45,46,47,48]
Alpha-2-HS-glycoprotein	FETUA	37	TGF-beta [49]
Apolipoprotein A-I	APOA1	31	Lipids, lipid receptors
Apolipoprotein A-IV	APOA4	45	Lipids, lipid receptors
Apolipoprotein B-100	APOB	509	Lipids, lipid receptors
Apolipoprotein E	APOE	36	Lipids, lipid receptors
Beta-2-glycoprotein 1	APOH	39	Phospholipids
Ceruloplasmin	CERU	121	Albumin (FcRn), Ctr1, ferritin, ferroportin [55,56,57,59,60]
Clusterin	CLUS	52	Low-density lipoprotein receptor
Complement factor I	CFAI	67	CD46 [64]
Hemopexin	HEMO	51	LRP1 [65]
Histidine-rich glycoprotein	HRG	59	Phospholipids [54]
Inter alpha-trypsin inhibitor, heavy chain 4	ITIH4	105	Hyaluronan
Plasminogen	PLMN	91	Apolipoproteins, thrombospondin
Pregnancy zone protein	PZP	166	LRP1, interleukin-1, Glycoprotein 78 [50,51,52,53]
Thrombospondin-1	TSP1	130	Integrins, LRP1, EGFR, TGF-beta, uPA, VEGF-A [66,67,68,69]

Abbreviations: number (#), vascular endothelial growth factor A (VEGF-A).

## Data Availability

Mass spectrometry data was uploaded for private access to the Mass Spectrometry Interactive Virtual Environment (MassIVE) database at https://massive.ucsd.edu/Pro-teoSAFe/private-dataset.jsp?task=c6b3c64e10184a298fd1e0f2477bec7f. The data is under the identifier MSV000090592 and can be accessed with the username (MSV000090592_reviewer) and password (Biomimetics).

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
