# Peer review of "Macromolecules Absorbed from Influenza Infection-Based Sera Modulate the Cellular Uptake of Polymeric Nanoparticles"

_biomimetics, 2022, doi:10.3390/biomimetics7040219_

Round 1

Reviewer 1 Report

In their manuscript entitled "Macromolecules absorbed from influenza infection-based sera modulate the cellular uptake of polymeric nanoparticles" the authors describe an very intersting study.

- The manuscript is very well written

- The results were presented very cleary, also the statistical analysis fulfill all requirements

- Expermental part is very well and can be followed by an expert in this field.

In summary it is a very good study, results are presented very clearly.

I recommend publication after spell check.

Author Response

Thank you for your comments.  We have conducted a thorough spell check using Grammarly and we should have caught the spelling errors.

Reviewer 2 Report

The Authors pre-treated HBPE-NPs by using sera from influenza A virus (IAV)-infected mice and this corona resulted in the increased accumulation of HBPE-84 NPs in tumors and reduced liver and spleen uptake. Overall these results support that the protein corona formed on polymeric NPs, like the HBPE-NPs pre-treated with sera from the IAV infection, is composed of unique proteins that modulate cancer or immune cell uptake to optimize the biological identity of polymeric NPs for targeted accumulation in tumors.

Minor

Fig 1: in order to better visualize the results it is suggested to keep the same fill for each graph bar for the same serum sample

Fig. 2: A. the overall quality of the images should be improved. In order to better visualize the results the magnification should be increased and the distribution of single images in the panel should be better organized. B. in order to better visualize the results it is suggested to keep the same fill for each graph bar for the same serum sample

Fig. 4: the overall quality of the images should be improved. In order to better visualize the results the magnification should be increased and the distribution of single images in the panel should be better organized.

Fig. 6: the overall quality of the image should be increased

The manuscript would need mild linguistic revision for typing errors

The Authors failed to give convincing evidences of their broad revision of the recent literature indeed they are still missing some important issues related to protein corona NPs decoration. Try to better outline the novelty of the here proposed strategy providing a broader and deeper discussion of the recent advances in the field.

Author Response

Thank you for your comments and suggestions.  Our response follows below.

  1. For Figure 1, the fill for each group in the graph has been adjusted as recommended by the reviewer.
  2. For Figure 2, the fill for each sera group was also adjusted as recommended by the reviewer. The images in the figure were reorganized to improve the presentation, and the image quality also improved.
  3. For Figure 4, the image quality was improved.  We did not change the distribution of single images since it now matches the distribution in Figure 2.  We also did not change the magnification since it is the same as in Figure 2. Improving image quality should help the reviewer better visualize the individual cells.
  4. The overall image quality was improved.  We are now including high-resolution images in the submitted files.
  5. The manuscript was reviewed using Grammarly to catch any typos or grammatical errors inadvertently introduced.
  6. We appreciate this comment and revised the Introduction and the Discussion to incorporate recent advances in the field that are relevant to our study of the protein corona on nanoparticles. A total of 11 new references were added to the manuscript.